# An Optoelectronics-Based Compressive Force Sensor with Scalable Sensitivity

**DOI:** 10.3390/s23146513

**Published:** 2023-07-19

**Authors:** Zachary Pennel, Michael McGeehan, Keat Ghee Ong

**Affiliations:** 1Department of Bioengineering, Knight Campus for Accelerating Scientific Impact, University of Oregon, Eugene, OR 97403, USA; zpennel@uoregon.edu (Z.P.); kgong@uoregon.edu (K.G.O.); 2Department of Physics, University of Oregon, Eugene, OR 97403, USA

**Keywords:** compressive force, optical sensors, elastomeric characterization

## Abstract

There is an increasing need to accurately measure compressive force for biomedical and industrial applications. However, this need has not been fully addressed, as many sensors are bulky, have high power requirements, and/or are susceptible to electromagnetic interference. This paper presents an optoelectronics-based force sensor that can overcome the limitations of many sensors in the market. The sensor uses a light emitting diode (LED) to transmit visible broad-spectrum light into a photoresistor through an optically clear spacer on top of an elastomeric medium. In the absence of an external force, the light path is mostly blocked by the opaque elastomeric medium. Under a compressive force, the clear spacer compresses the elastomer, moving itself into the light path, and thus increasing the overall light transmission. The amount of light received by the photoresistor is used to quantify compressive force based on elastomer displacement/compression and a priori knowledge of elastomer stiffness. This sensing scheme was tested under eight different configurations: two different sized sensors with four types of elastomers per size (20A neoprene, 30A neoprene, 50A neoprene, and 75A styrene–butadiene rubber (SBR)). All configurations measured force with R^2^ > 0.97, RMSE < 1.9 N, and sensitivity values ranging from 17 to 485 N/V. This sensing scheme provides a low-cost, low-power method for accurate force sensing with a wide force range.

## 1. Introduction

Force sensors are used in industrial processes [1,2,3], consumer products, and healthcare fields such as sports medicine [4,5,6]. For example, force sensors are used to provide biofeedback for customizing rehabilitation therapy in sports medicine [7,8,9,10,11]. These sensors are also used congruently with cameras, gyroscopes, and other sensors to assist autonomous and semi-autonomous robots in sensing and interacting with their surroundings [12,13]. While a wide variety of force sensors are available commercially, many have narrow force ranges, thus requiring multiple distinct types of sensors when monitoring a broad force range [14,15,16]. Furthermore, many force sensors are susceptible to electromagnetic interference (EMI), which is increasingly problematic as technological advancements result in smaller electronic components. For example, capacitive-based force and pressure sensors, which have been used for decades, are exposed to greater EMI as they are placed closer to other signal-carrying components [17]. In addition to EMI, resistive and capacitive sensors also draw large amounts of power when in use [18,19], which affect their practical utility as embedded and wireless sensors. As such, there remains a need for a low-cost, low-power force sensor with scalable dimensions that can sense a broad range of forces and remain unaffected by EMI. Such a sensor would have broad uses in wearable and embedded applications such as biomechanical monitoring, industrial manufacturing, and robotics.

Optical approaches have been successfully implemented for measuring multi-axial shear forces in wearable devices [20,21,22]. It was demonstrated that shear force can be accurately measured using an LED and a photoresistor. In this paper, we present an optical-based approach for measuring compressive (i.e., normal) forces. The force-sensing range and sensitivity can be tuned by varying the material properties of the internal elastomer. Elastomers often exhibit nonlinear stiffness (stress vs. strain) responses [23]. Since our sensor is based on changes in light due to the deformation of the elastomeric medium, our sensor response is similarly nonlinear. 

This paper presents the design, fabrication, and characterization of an optical-based compressive force sensor. The sensor has unique properties that make it advantageous for scalability, size, and measuring forces over a broad range. The sensor exhibits a nonlinear stiffness response, allowing it to operate under a broad range of forces; resolution is higher at low forces while avoiding saturation at higher forces. The interchangeability of elastomers and dynamic resolution at different force ranges make the sensor easily scalable and highly versatile. These properties, combined with a small form factor and low power requirements, make this sensor well-suited for force sensing for a variety of applications such as healthcare monitoring, industrial manufacturing, and robotics.

## 2. Materials and Methods

### 2.1. Design

The operational principle of the sensor is based on changes in light transmitted from an LED to a photoresistor (Figure 1). A voltage divider circuit (Figure 1c) is used to convert changes in resistance across the photoresistor to voltage variations, which are then digitized and recorded by a microcontroller (MCU) through a built-in 10-bit analog-to-digital converter. Compressive force measurements are achieved by incorporating an optically clear spacer on top of a compressible light-absorbing elastomer between the LED and photoresistor. As force is applied to the elastomer, it is compressed, displacing more of the clear spacer into the light-transmission path. This increases light transmission between the LED and photodiode, and thus the voltage measured by the MCU (Figure 1a,b).

The sensor comprises three primary components: the electronics enclosed in a polylactic acid (PLA) housing, a transparent spacer, and an elastomer (Figure 2). The PLA housing was 3D-printed with a 1.75 mm filament printer (Original Prusa i3 MK3S+, Prusa Research, Prague, Czech Republic), and the transparent plate was fabricated with Formlabs Clear v4 photopolymer resin using a stereolithography (SLA) 3D printer (Form 3+, Formlabs, Somerville, MA, USA). The sensor electronics consist of a 5 mm common anode RGB LED (Adafruit 2739, Adafruit Industries LLC, New York City, NY, USA) placed adjacent to a cylindrical well and directly facing a photoresistor (Adafruit 161, Adafruit Industries LLC, New York City, NY, USA). Two sizes of well were investigated: A 4.8 mm deep well with a 6 mm diameter (see the high-force sensor below), and a 10 mm deep well with a 12 mm diameter (see the low-force sensor below). This specific RGB LED was chosen for its minimal power consumption and spectral controllability, which may be an important feature in future work (e.g., the potential to integrate with an optical-based shear sensor) [24]. To control the amount of light transmission, a 3.25 mm optically opaque elastomer was placed at the bottom of the well, partially covering both the LED and photoresistor as shown in Figure 2. As a force is applied to the top of the transparent spacer, it compresses the elastomer, allowing increased light transmission from the LED to the photoresistor and decreasing resistance across the voltage divider circuit. The relationships between applied force, elastomer compression, and light transmission are dependent upon the material properties of the elastomeric medium.

Two sizes of sensors were fabricated and characterized to demonstrate the scalability and versatility of this technology: (a) high-force sensors that are designed for broader force sensing, and (b) low-force sensors that have higher sensitivity and smaller form factors (Figure 2). The high-force sensor has a nominal height of 17.6 mm, while the low-force sensor has a height of 9.0 mm. The force-sensing ranges of both types of sensors can be adjusted by using elastomers with different moduli or altering their elastomer dimensions.

### 2.2. Sensor Characterization

The compressive force response of each sensor was characterized with four Shore A durometer elastomeric media: 20A neoprene, 30A neoprene, 50A neoprene, and 75A styrene–butadiene rubber (SBR). Material characterization was performed using a material testing system (MTS) (Electroforce 3200, TA Instruments, New Castle, DE, USA). A circular punch was used to cut each elastomer into 6 mm or 12 mm diameter disks for the low- or high-force sensors, respectively. All elastomeric disks were 3.25 mm ± 0.04 mm thick. The stiffness properties of each elastomer type and size were independently characterized to establish the baseline relationships between load and displacement and serve to elucidate the predicted relationship between the mechanical properties of the sensor and its ability to measure force (Section 2.2.1). The sensor’s ability to accurately measure compressive force was characterized (Section 2.2.2). The hysteresis of the sensor was measured to evaluate accuracy during loading and unloading conditions. Lastly, fatigue tests were performed to evaluate the sensor’s ability to accurately measure force with repeated use. Mechanical stiffness of the sensor was also measured at pre- and post-fatigue procedures.

#### 2.2.1. Elastomer Characterization Methods

Stiffness characterization tests were performed for two different elastomer materials: neoprene (20A, 30A, and 50A) and SBR (75A). For the large-force sensors, elastomer characterization was performed on the MTS with a continuous triangle force waveform at 0.1 Hz duty cycle for 5 cycles with a force range of 1 N (pre-load) to 100 N, which is the maximum testable load on the MTS. In contrast, the small force sensors were characterized under a 0.1 Hz continuous triangle waveform for 5 cycles with 1 N (pre-load) to an upper force limit that was dependent on the elastomer to ensure compression never exceeded 50% of elastomer thickness. Testing the small elastomer samples to a 50% compression threshold was necessitated by the inability of these samples to reach the 100 N threshold without bottoming out. The waveform frequency was chosen based on quasi-static loading and typical human walking speed (~2 Hz) [25]. Displacement and force were recorded at a frequency of 10 Hz for the duration of the tests. For all tests, loads were applied along the central axis and distributed uniformly across the cylindrical base.

#### 2.2.2. Sensor Characterization Methods

Housing units for the low- and high-force sensors were designed to be mounted onto the MTS for testing (Figure 3). Two mounting platforms were 3D-printed (Original Prusa i3 MK3S+, Prusa Research, Prague, Czech Republic) using 1.75 mm PLA filament for the sensor’s clear spacer and housing (Figure 3a,d). Two 10–32 machined bolts were used to secure each sensor to the mounting hardware (Figure 3a), which was connected directly to the MTS (Figure 4a). The high-force sensor’s clear spacer was attached to the other mounting hardware with two 6-32 bolts, and then attached directly to the MTS. The low-force sensor mounting hardware had a similar attachment method to the MTS. The housing of the low-force sensor was attached to the mounting hardware via two perpendicularly oriented 6-32 bolts (Figure 3d and Figure 4b).

During testing, the sensor was connected to a 10-bit MCU controlled via a MATLAB script that recorded voltages (accuracy: ±5 mV) across the photoresistor and timestamps. The MTS was preprogrammed to perform a force-controlled compression test using a proportion–integral–derivative (PID) feedback controller (low-force sensor: 5 cycles at 0.1 Hz (1 N–25% displacement or 100 N); high-force sensor: 5 cycles at 0.1 Hz (1–100 N)). For the low-force sensors, a displacement limit of 25% elastomer thickness was imposed to avoid sensor damage. This means that each elastomer was compressed to either 25% thickness or 100 N, whichever condition was met first. The MTS recorded local timestamps and force from an in-series load cell (1516FQG-100, TA Instruments, New Castle, DE, USA) (accuracy: ±0.01 N), and displacement from an in-series high-accuracy displacement sensor (accuracy: ±0.0001 mm). The measurements collected by the MTS were synchronized with the MCU data via their respective timestamps (see Section 2.3).

During the hysteresis test, the low-force sensors were subjected to 5 loading/unloading cycles at 0.1 Hz duty cycle (1 N–25% displacement or 100 N), while the high-force sensors were subjected to 5 cycles at 0.1 Hz (1–100 N). Both types of sensors were tested with the MTS using a triangular waveform consisting of a compressive and a decompressive phase. Fatigue testing (10,000 cycles at 1 Hz duty cycle) was performed for each sensor configuration. Stiffness of each configuration was evaluated (low-force sensor: 0–30 N; high-force sensor: 0–100 N) before and after the fatigue tests. There were no displacement limits for hysteresis tests to ensure that degradation was performed under consistent compressive force. This was motivated by the tendency of elastomer mechanical properties to change with cyclical loading due to the breaking of chemical bonds and heat [26,27].

### 2.3. Data Analysis

Custom MATLAB scripts were used to analyze the collected data. In order to accurately synchronize sensor data and the MTS record, all sensor data were interpolated linearly to match the length and time steps of the MTS record. Both sensor and MTS data records were averaged over 5 cycles, and standard deviations were calculated at each data point.

The stiffness of the material is defined as the ratio of change in load over the displacement. Each elastomer configuration was characterized across five loading trials, and the mean and standard deviation of the response were calculated. Sensor data were analyzed to determine their sensitivity, defined as change in applied force over the measure change in voltage. A second-order polynomial function was used to model the sensor’s sensitivity curve. To visualize the biphasic response, two linear piecewise functions were calculated using linear regression with least squares fitting of a given subset of data. Data were partitioned at the point that minimized the squared error of each region from its local mean. The MTS-measured force was then compared to the sensor-estimated force based on the polynomial regression to determine sensor performance, correlation coefficient (R^2^), the sum of squares error (SSE), and root-mean-squared error (RMSE) values.

Hysteresis was calculated as the ratio of the difference between the compression and decompression conditions at the midpoint of the force range and the difference between the minimum and maximum displacement values. Percentage degradation from the fatigue test was calculated as the percentage change in average stiffness across the loading range before and after the 10,000-duty-cycle fatigue protocol (Section 2.2.2).

## 3. Results and Discussion

### 3.1. Material Charactization

Figure 5 depicts the load-displacement responses of the sensors with different elastomers. As shown in the figure, sensors with 6 mm diameter elastomers (right column) exhibited lower stiffness compared to their 12 mm diameter counterparts (left column). Sensors with 6 mm elastomers also showed a two-phase stiffness response, with an initially steep slope that tapered off at approximately 50% of the maximum load. This response is exemplified in sensors with 6 mm 20A neoprene elastomers (Figure 5b).

Overall, the smaller elastomers (Figure 5b,d,f,h) exhibited lower stiffness than the larger elastomers (Figure 5a,c,e,g). The small sensors with 20A and 30A elastomers showed a two-phase linear relationship, but the two-phase behaviors disappeared as the stiffnesses of the elastomers increased. As expected, the 75A durometer samples had the lowest variation throughout the loading range. The 12 mm 75A sample was the stiffest elastomer tested (1.33 kN/mm) and had the lowest standard deviation (SD) (SD < 0.11% across the full loading range). In contrast, the 6 mm 20A sample was the least stiff elastomer tested (20.5 N/mm) with the largest standard deviation of all 6 mm samples (SD < 2.34% across the full loading range).

### 3.2. Sensor Response

Sensor responses are dependent on the elastomer material inside the well of the sensor housing. By changing the size and elasticity of the elastomer, the sensor sensitivity was tuned from 17 N/V (20A, 6 mm) to 485 N/V (75A, 12 mm). For most elastomer configurations, the sensor exhibited a two-phase response (a low-force response followed by a high-force response), with the latter phase having a decreased force sensitivity.

Sensors with 20A durometer neoprene elastomers had the lowest force-sensing range of all configurations tested while having the highest sensitivity (Figure 6a,b). Sensor-derived force measurements based on the second-order polynomial fit matched the load cell data well (R^2^ > 0.99, SSE = 34.71 N, RMSE = 0.94 N). The low-force 20A sensor had a similar accuracy performance (R^2^ > 0.99, SSE = 0.41 N, RMSE = 0.103 N). For both configurations, the sensor variations scaled approximately with the force magnitude (Figure 6c). This may be explained in part by the flexing of the sensor housing under higher loads, as previous research has shown higher variation in elastomer mechanical behavior under greater stress [28].

Sensors with 30A neoprene elastomers (Figure 7a,b) showed similar force sensitivity profiles to those with 20A neoprene elastomers. Specifically, the 30A high-force configuration demonstrated good agreement with the load cell data (R^2^ > 0.99, SSE = 17.95 N, RMSE = 0.683 N). The low-force sensor had similar levels of agreement (R^2^ > 0.99, SSE = 0.11 N, RMSE = 0.05 N). Variation was consistent for both configurations, suggesting that error may have occurred due to inherent material softening, as described by the Mullin’s effect for vulcanized rubber elastomers where softening occurs following recurring stresses lower than or equal to the experimental maximum stress applied [29]. The high-force sensitivity curve was more linear than the low-force curve, with percentage change in the second phases being 123% and 72% for high- and low-force configurations, respectively. Material characterization of the high-force 30A neoprene sensor showed a nonlinear stiffness profile (Figure 5c,d), which indicates that the light-sensing paradigm may exhibit a nonlinear response under the load range tested. Future work should seek to characterize the sensor response across a broader range of forces.

Results from sensors with 50A neoprene elastomers (Figure 8a,b) showed their ability to measure the largest force ranges compared to the other neoprene configurations. The high-force sensors demonstrated good levels of agreement (R^2^ > 0.99, SSE = 99.64 N, RMSE = 1.88 N). The low-force configuration also matched the load cell data (R^2^ > 0.99, SSE = 1.17 N, RMSE = 0.20 N). Predicted force for the high-force configuration began deviating from the true force near 65 N. Like the 20A neoprene, this may be due to flexing of the sensor housing or increased variability in elastomer mechanics at higher forces [28]. The high-force sensor response (Figure 8a) for the 50A neoprene was also more linear than the material stiffness characterization test (Figure 5e). This discrepancy could be indicative of nonlinear change in incident light with increasing compressive force for the range tested. Inversely, the sensor response of the low-force configuration (Figure 8b) was nonlinear.

Sensors with 75A SBR elastomers had the highest force-sensing ranges of the sensors tested (Figure 9a,b). For the high-force sensor, the predicted force closely aligned with load cell data (R^2^ > 0.99, SSE = 14.42 N, RMSE = 0.71 N). The 75A elastomer configuration had the lowest and most consistent predicted force error of all high-force sensors tested. The low-force sensor followed less closely to the model than the high-force sensor (R^2^ > 0.98, SSE = 51.62 N, RMSE = 1.34 N). Similar to the neoprene configurations, residuals were greatest under higher forces [28]. The low-force configuration residuals (Figure 9f) had distinguishable drift that increased with each loading cycle [29,30]. Overall voltage changes in the 75A high-force sensor (<0.25 V across the full loading range) were the smallest of all configurations tested. Lower light intensity correlated to lower displacement and model error, which is indicative that lower displacement values are associated with a decreased error in force prediction. The low-force 75A configuration had the broadest force sensitivity and most associated error out of all low-force sensors tested; this is most likely due to slower stress relaxation from material stiffness causing drift after each compressive cycle.

For most sensors, the highest sensing error occurred for loads near 100 N, which was the limit of the test due to the capacities of the MTS and load cell. Sources of error in this range may include changes in material properties (e.g., Mullin’s effect) throughout the five-cycle test that caused the response curves of the materials to drift over time. This pattern can be seen in the residual plots of many of the tested elastomer types irrespective of size. It should be noted that the drift of the 6 mm elastomers was less than that of the 12 mm elastomers. Drift may also be attributed to the inherent properties of elastic materials, such as temperature-dependent variations in mechanical properties [30]. It is also possible that the PLA sensor housing began to flex at higher loads, which would not be captured by the sensor and thus manifest as error. In this study, we only sought to model the sensor’s behavior under a compressive force; however, future work should also evaluate sensor behavior under decompression or tensile loads. Similarly, future work should seek to characterize the accuracy and error profile of the sensor across a larger force range.

Results suggest that a sensor configured with 20A neoprene would be the most suitable for applications requiring higher sensitivity with low force range. Inversely, for broader force ranging capability, higher durometer materials such as the 75A SBR tested would be the most useful. Both 30A configurations exhibited the lowest fatigue-induced degradation and hysteresis. These results indicate that 30A neoprene would perform best under conditions with cyclical loading with relatively consistent amplitude profiles.

### 3.3. Sensor Hysteresis

Overall, hysteresis was lower in the low-force sensors compared to the high-force sensors (Figure 10). The 20A low-force sensor had the lowest hysteresis of all low-force configurations tested (high-force = 8.67%, low-force = 3.58%). The 30A sensors had similar hysteresis values, with exception to the high-force configuration, which had the lowest hysteresis of all high-force configurations (high-force = 6.25%, low-force = 5.54%). The 50A sensors had moderately large hysteresis values (high-force = 16.79%, low-force = 16.38%). The 75A sensors had the largest hysteresis of all sensors tested (high-force = 18.80%, low-force = 21.41%).

Sensors with higher durometer elastomers showed more hysteresis compared to sensors with elastomers of lower durometers, with the exception of the 20A high-force configuration (Figure 10). This response matches previous research which has shown high-durometer elastomers to exhibit greater hysteresis than low-durometer elastomers [31]. Durometer–hysteresis correlation is found at the molecular level, as higher durometer rubbers are toughened by fillers, resulting in a stiffer elastomer with larger hysteresis. Similarly, low-force sensors had lower hysteresis due to lower energy dissipation in the elastomers during compression [29].

### 3.4. Fatigue Characterization

It was found that the 20A high-force module had a degradation (i.e., reduction in mechanical stiffness) of 14.1%. The 30A configuration degraded by 11.7%, the least out of all neoprene durometers. The most degraded sensor configuration was the 50A neoprene, which degraded 14.6%. In contrast, the 75A SBR degraded by only 4.4%. The 20A low-force module had a degradation of 1.07%. The 30A configuration had a similar degradation of 1.06%. The 50A configuration degraded 0.91%, the least of all neoprene configurations. The 75A SBR configuration degraded 0.09%, the least of all configurations tested. These data indicate that, with prolonged use, periodic recalibration of the sensor may be required to account for elastomer degradation. This is especially relevant for the neoprene configurations, which showed the most degradation.

### 3.5. Sensor Application

The primary goals of this work were to create a compressive force sensor that has a small form factor (size), low cost, low power requirements, and resistance to electromagnetic interference. Secondary goals were to develop a scalable and versatile sensing paradigm capable of measuring a broad range of loads and load characteristics. Through mechanical testing and characterization of eight sensor configurations, we have shown this optical compressive sensing paradigm to be scalable and tunable for a variety of sensing parameters. The low- and high-force sensor configurations weigh 1.96 g and 7.22 g, respectively, which are comparatively lower than many other force sensors previously reported in the literature (e.g., Liu et al. (2009) [16]). Furthermore, this sensor combines a wireless design, small size, low mass, and low power requirements into a single package that may be advantageous compared to other sensors which are limited to only some of these criteria (Table 1) [3,14,16,32,33]. For example, Ueda et al. (2007) [32] employed an optical-based design whereby a high-speed camera was used to measure the diffraction of light through an acrylate layer. This method is low-powered but requires a bulky camera to function in an enclosed environment. Another optics-based force sensor uses a photodetector and small threaded optical fibers [33], which are more costly and difficult to assemble than the sensor components presented here. Many other force sensing concepts rely on either capacitive sensing or strain gauge technology which typically have narrow force-range sensing capabilities, require greater power, and are sensitive to EMI [14,15].

Variations in sensitivity and operational force range based on elastomer material and durometer suggest that other elastic materials (e.g., polymers and springs) could also be implemented in future deigns to expand the utility of this sensor and sensing paradigm for different force measurement applications. Ethylene propylene diene monomer (EPDM) rubber, silicone, and chloroprene may be useful elastomers due to their well-studied material properties and inexpensive manufacturing costs. While this work has tested a broad range of Shore A durometer materials, it did not include the ends of the Shore A classification—0A and 100A—and thus did not yield the highest and lowest sensitivity curves, respectively [30]. Future work should seek to confirm this. Furthermore, all sensor tests in this study were performed at a consistent loading rate under room temperature conditions. Future work will aim to evaluate temperature- and loading-rate-dependent sensor responses.

A unique property of the force-sensing paradigm presented in this paper is the two-phase sensitivity response, whereby the sensors exhibit higher sensitivity and precision for small force measurements yet maintain the ability to measure higher forces where sensitivity and precision are relatively less important. This sensing paradigm could be especially useful in applications such as continuous tracking of biomechanical activities, an application which requires measuring broad force ranges (e.g., standing, walking, running, and jumping). For instance, during a low intensity activity such as walking, hip flexion and extension forces are <300 N, hip external rotation is <200 N, and ankle elevation is <1000 N but could increase by a factor of 5 with highly dynamic activity [34,35,36,37,38].

All mechanical tests of materials were performed via continuous force-controlled loading. Previous tests showed drifting of sensor-derived force measurements when under quasi-static loading conditions, which increased with higher forces. Relaxation under quasi-static loading is likely due to inherent properties of the viscoelastic materials chosen for this study [30,39]. Future research may seek to characterize this response and compensate for it through material selection or computational approaches. Future efforts will also focus on decreasing the size of the sensor module. This can be accomplished by sourcing smaller LEDs and photoresistors. The dimensions of the components used in this study dictated sensor volume. The relatively bulky LED (5 mm × 4.9 mm × 2.4 mm) constrained displacement distance and minimum size of the sensor housing. The overall size was also constrained by limits of properly mounting the sensor to the MTS for testing. For smaller-form-factor sensors, a new testing approach would need to be implemented to successfully characterize them. 

The goal of this study was to develop a novel sensor and demonstrate its scalability by varying elastomer type (neoprene and SBR), durometer, and size. Future work will seek to implement and evaluate this sensor for practical analytical applications. Specifically, next steps include combining the compressive force sensor with a two-axis optical shear force sensor [20] for complete three-axis force measurements. Future work will seek to integrate the three-axis force sensor into footwear for continuous gait biomechanics monitoring.

## 4. Conclusions

This paper presents a novel, compressive force sensor based on an LED and photoresistor, as well as a compressible elastomer that blocks the LED light in response to an applied force. This sensor is scalable, low cost, and low weight. Eight different embodiments were tested, illustrating the scalability and repeatability of the concept. All sensor configurations exhibited strong relationships between load and voltage (i.e., light intensity) modeled by a second-order polynomial fit (R^2^ > 0.97 for all eight configurations). These sensors can measure compressive force of up to 100 N, with sensitivity values ranging 17–485 N/V, exemplifying the scalability and versatility of the design. These parameters may be expanded upon in future iterations by varying the size and material within the sensor. The performance and tunability of the sensor support its use for a wide variety of biomedical applications and robotics.

## Figures and Tables

**Figure 1 sensors-23-06513-f001:**
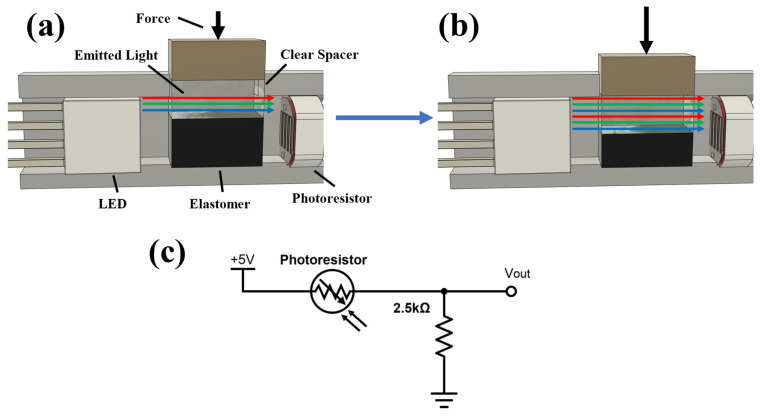
Illustration of the sensing paradigm. (**a**) Resting (i.e., zero force) condition, (**b**) increase in light transmission to the photoresistor under a compression force, and (**c**) circuit diagram of the voltage divider that converts resistance of the photoresistor into voltage.

**Figure 2 sensors-23-06513-f002:**
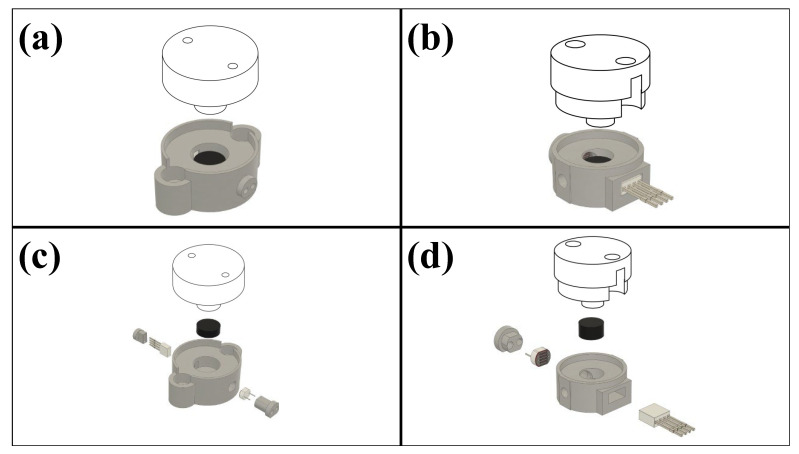
Renderings of a fully assembled high-force sensor (30 mm outer diameter) (**a**) and a low-force sensor (15 mm outer diameter) (**b**). Exploded rendering of the high-force (**c**) and low-force (**d**) sensors.

**Figure 3 sensors-23-06513-f003:**
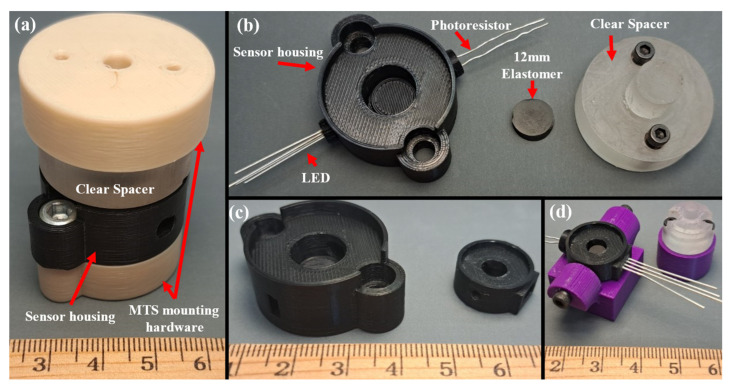
Photographs of the sensors. (**a**) An assembled high-force sensor (excluding LED and photoresistor) for MTS testing. (**b**) A partially assembled high-force sensor. (**c**) Comparison of the high- and low-force sensors (**d**) An assembled low-force sensor set up for MTS testing.

**Figure 4 sensors-23-06513-f004:**
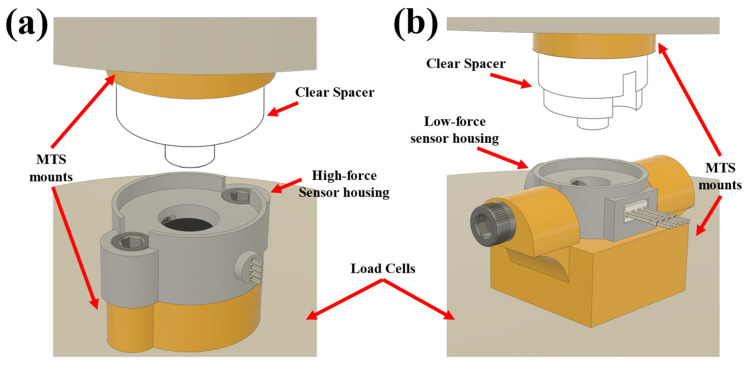
Drawings depicting the MTS setup for characterizing (**a**) high-force sensors and (**b**) low-force sensors.

**Figure 5 sensors-23-06513-f005:**
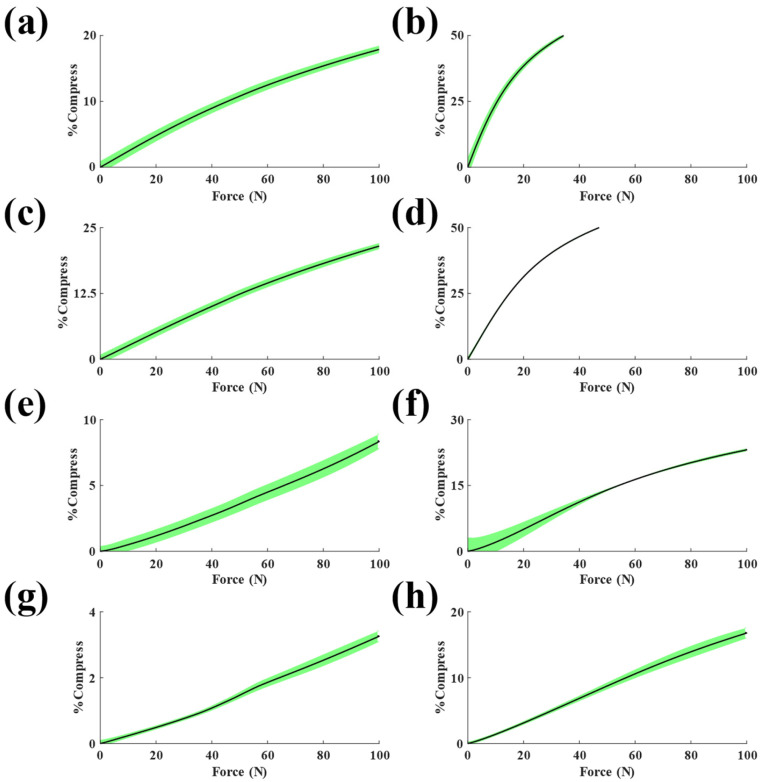
Force versus displacement plots for sensors with 12 mm (**left** column) and 6 mm (**right** column) diameter elastomers. The types of elastomers used in the experiments were 20A (**a**,**b**), 30A (**c**,**d**), 50A (**e**,**f**), and 75A (**g**,**h**). Data are mean ± standard deviation for *n* = 5 loading cycles.

**Figure 6 sensors-23-06513-f006:**
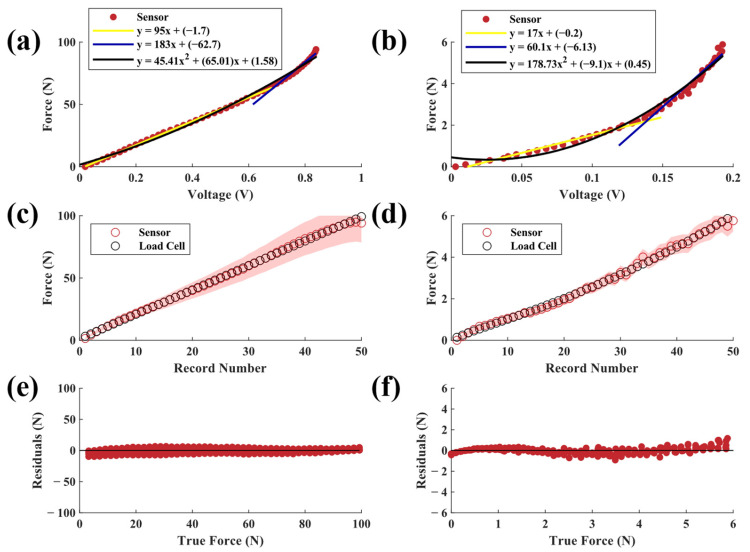
Applied force versus output voltages from (**a**) high-force and (**b**) low-force sensors with 20A neoprene elastomers. Each plot also shows two piecewise linear fits and a second-order polynomial fit. Predicted (mean ± SD) vs. actual force for (**c**) high-force and (**d**) low-force sensors. Raw residuals of predicted force vs. true force for (**e**) high-force and (**f**) low-force sensors.

**Figure 7 sensors-23-06513-f007:**
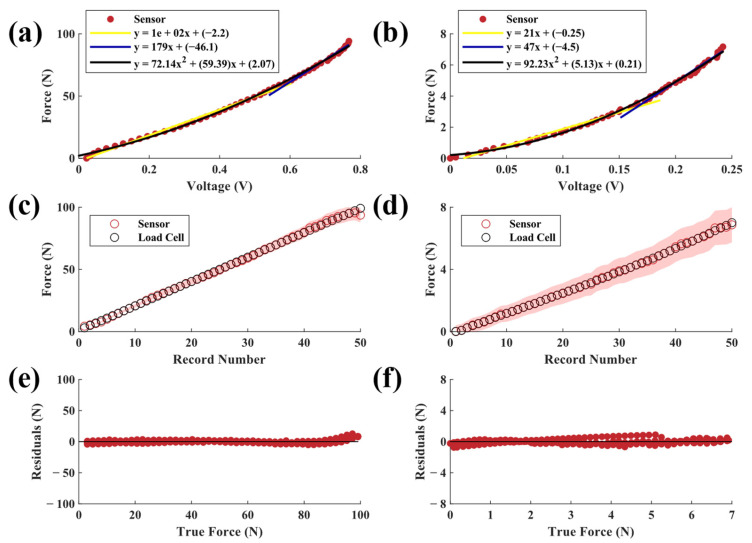
Applied force versus output voltages from (**a**) high-force and (**b**) low-force sensors with 30A neoprene elastomers. Each plot also shows two piecewise linear fits and a second-order polynomial fit. Predicted (mean ± SD) vs. actual force for (**c**) high-force and (**d**) low-force sensors. Raw residuals of predicted force vs. true force for (**e**) high-force and (**f**) low-force sensors.

**Figure 8 sensors-23-06513-f008:**
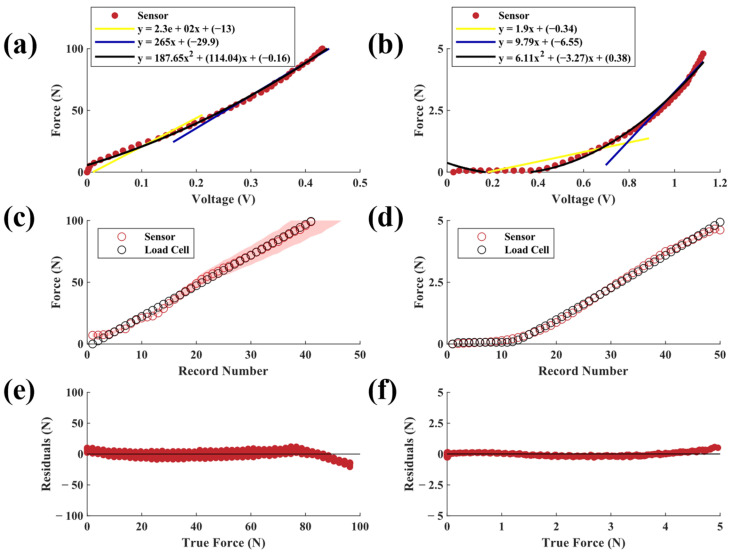
Applied force versus output voltages from (**a**) high-force and (**b**) low-force sensors with 50A neoprene elastomers. Each plot also shows two piecewise linear fits and a second-order polynomial fit. Predicted (mean ± SD) vs. actual force for (**c**) high-force and (**d**) low-force sensors. Raw residuals of predicted force vs. true force for (**e**) high-force and (**f**) low-force sensors.

**Figure 9 sensors-23-06513-f009:**
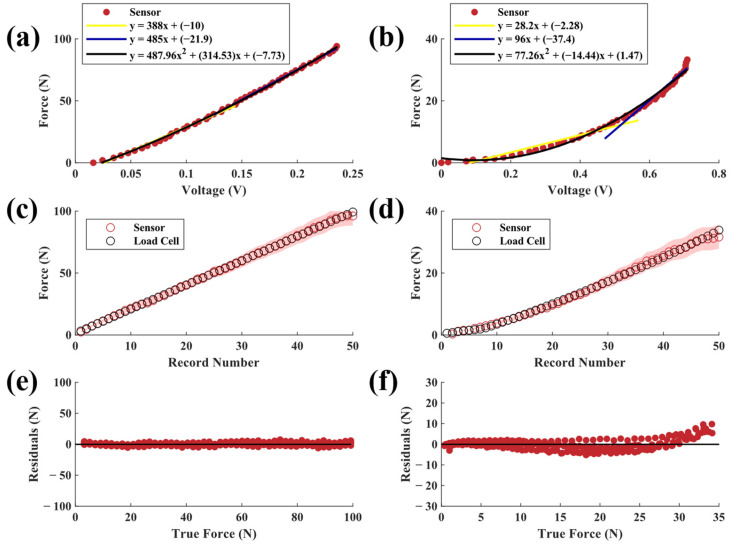
Applied force versus output voltages from (**a**) high-force and (**b**) low-force sensors with 75A neoprene elastomers. Each plot also shows two piecewise linear fits and a second-order polynomial fit. (**c**,**d**) Predicted (mean ± SD) vs. actual force for (**c**) high-force and (**d**) low-force sensors. (**e**,**f**) Raw residuals of predicted force vs. true force for (**e**) high-force and (**f**) low-force sensors.

**Figure 10 sensors-23-06513-f010:**
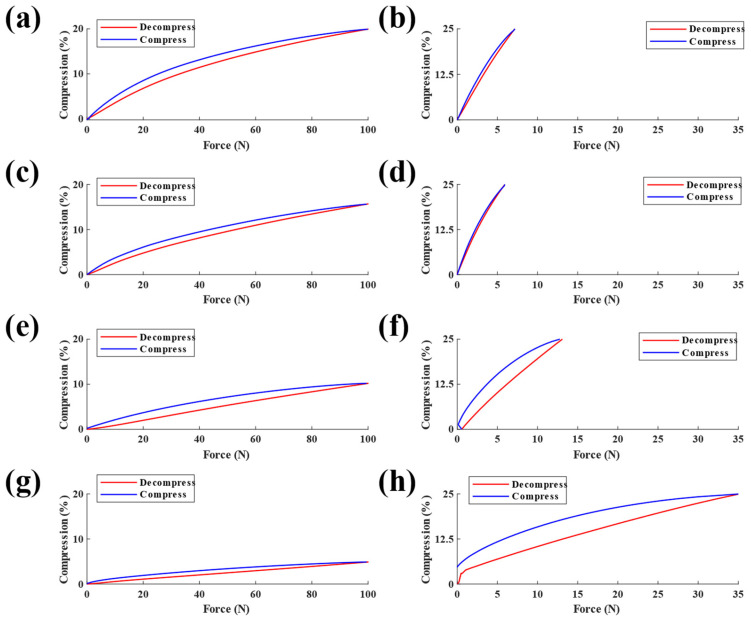
Hysteresis of sensor responses for low-force sensors with 20A, 30A, 50A, and 75A elastomers, respectively (**a**,**c**,**e**,**g**) and high-force sensors with the same elastomers (**b**,**d**,**f**,**h**).

**Table 1 sensors-23-06513-t001:** Comparison of other similar sensors with key design parameters.

Study	Sensing Principle	Mass	Diameter	Power Source	Sensing Capacity
Present Study	Optoelectronic (Photoresistor and LED)	7.22 g 1.96 g	30 mm 15 mm	Wired/Wireless (30 mA)	Normal Force: 100 N 30 N
Bodini et al. (2018) [14]	Capacitive	*	8.5 mm	Wired	Normal Force: 1 N
Liu et al. (2016) [15]	Resistance Strain Gauge (variable resistor)	*	9.62 mm	Wired (11 mA)	Normal Force: <1 N
Liu et al. (2009) [16]	Pressure sensitive electric conductive rubber (PSECR)	10g	10 mm	Wired	Normal Force: 100 N Shear Force: 35 N
Ueda et al. (2007) [32]	Optical/Camera	*	240 mm	Wired	Normal Force: 60 N
Avellar et al. (2021) [33]	Optoelectronic (LED and Photodetector)	*	*	Wired	Normal Force: 60 N

* Data not reported.

## Data Availability

Data will be made available upon reasonable request to the authors.

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
