# Peer review of "An Optoelectronics-Based Compressive Force Sensor with Scalable Sensitivity"

_sensors, 2023, doi:10.3390/s23146513_

Round 1

Reviewer 1 Report

In this manuscript, the authors reported a sensitive optoelectronics-based compressive force sensor. Some interesting results are obtained. I, therefore, recommend an acceptance for publishing after the next revisions.

Please provide information about the original aspect of the article at the end of the introduction.

Comment on the studies given in Table 1 and indicate the superior aspects of the given study.

The resolution of the figures is very low and not possible to read. Please supply the high resolution.

There are errors in punctuation throughout the article. There should be a space before the references given in the text.

Reviewer 2 Report

The manuscript is well written, and well organized. However, I suggest the following comments before in publication.

1) Please improve the introduction section (add the novelty of your work and importance of your sensor)

2) Have the authors investigated the effect of elastomer type?

3) Has the effect of environmental parameters such as temperature, etc. on the performance of the sensor been investigated?

4) For better comparison, please add your study in Table 1. 

5) Have the authors used the proposed sensor for practical analytical purposes, such as the detection and measurement of biomaterials

Reviewer 3 Report

This paper presents an optical-based approach for measuring compressive forces and the force sensing range and sensitivity can be tuned by varying the material properties of the internal elastomer. The advantage of the proposed designs is the nonlinear stiffness response, that allows the sensor to operate under a broad range of forces.

The manuscript is well written and presented. The topic is in line with current interest of researches. The introduction well present the problem and introduce the aim of the research. The design and the characteristics of the sensors are well described and supported by graphical representations. The data analysis in correctly presented. Results are well reported and discussed. I suggest to sum up the limitations and the future works at the end of the manuscript, in order to underline next steps of the research. Some preliminary results of the use of the sensor in realistic application may be added.

Minor editing of English language required
